# Pyrazinamide and derivatives block ethylene biosynthesis by inhibiting ACC oxidase

Xiangzhong Sun[1,2,3,4,*], Yaxin Li[1,*], Wenrong He[1,5], Chenggong Ji[1], Peixue Xia[1,2,3], Yichuan Wang[4], Shuo Du[1], Hongjiang Li[1,5], Natasha Raikhel[5], Junyu Xiao[1,3] & Hongwei Guo[3,4]

Ethylene is an important phytohormone that promotes the ripening of fruits and senescence of flowers thereby reducing their shelf lives. Specific ethylene biosynthesis inhibitors would help to decrease postharvest loss. Here, we identify pyrazinamide (PZA), a clinical drug used to treat tuberculosis, as an inhibitor of ethylene biosynthesis in *Arabidopsis thaliana*, using a chemical genetics approach. PZA is converted to pyrazinecarboxylic acid (POA) in plant cells, suppressing the activity of 1-aminocyclopropane-1-carboxylic acid oxidase (ACO), the enzyme catalysing the final step of ethylene formation. The crystal structures of *Arabidopsis* ACO2 in complex with POA or 2-Picolinic Acid (2-PA), a POA-related compound, reveal that POA/2-PA bind at the active site of ACO, preventing the enzyme from interacting with its natural substrates. Our work suggests that PZA and its derivatives may be promising regulators of plant metabolism, in particular ethylene biosynthesis.

[1] The State Key Laboratory of Protein and Plant Gene Research, School of Life Sciences, Peking University, Beijing 100871, China. [2] Academy for Advanced Interdisciplinary Studies, Peking University, Beijing 100871, China. [3] Peking-Tsinghua Center for Life Sciences, Beijing 100871, China. [4] Department of Biology, Southern University of Science and Technology, Shenzhen, Guangdong 518055, China. [5] Center for Plant Cell Biology, Department of Botany and Plant Sciences, University of California, Riverside, California 92507, USA. * These authors contributed equally to this work. Correspondence and requests for materials should be addressed to J.X. (email: junyuxiao@pku.edu.cn) or to H.G. (email: guohw@sustc.edu.cn).

Ethylene gas is a vital plant hormone with diverse functions in plant growth and development, defense response, as well as adaption to stress. It has unique value both in basic research of plant hormones and agricultural practice. Over the past three decades, with the help of genetic analysis and biochemical approaches, a relatively linear signalling pathway of ethylene has been established[1]. Ethylene gas binds to its receptors, a class of histidine-like kinase located on the endoplasmic reticulum membrane, results in the inhibition of CONSTITUTIVE RESPONSE1 (CTR1), a Raf-like Ser/Thr protein kinase that acts as a negative regulator of ethylene signalling[1]. Upon CTR1 inactivation, one of its substrates, ETHYLENE-INSENSITIVE 2 (EIN2), becomes unphosphorylated and is subsequently subjected to site-specific proteolysis[2]. The C-terminal cleavage fragment of EIN2 is then translocated to the nucleus to activate the master transcription factor ETHYLENE-INSENSITIVE 3 (EIN3) and its homologue EIN3-Like 1 (EIL1)[1,2].

The ethylene biosynthesis pathway in higher plants is well characterized. In short, ethylene is synthesized from S-adenosylmethionine (SAM), which is converted to 1-aminocyclopropane-1-carboxylate (ACC) by the enzyme ACC synthase (ACS). ACC is then oxidized by the ACC oxidase (ACCO or ACO, referred to as ACO in this study), giving rise to ethylene, carbon dioxide ($CO_2$) and cyanide[3]. Usually, plant ethylene production is maintained at a low basal level, but is induced rapidly and dramatically under certain developmental stages or stress stimulations[3,4]. Although ACS is generally considered as the rate-limiting step in ethylene biosynthesis, there is growing evidence that ACO acts also as a control point under specific developmental and stress conditions in various plant species[5–7].

ACO enzyme is a 2OG-oxygenase 'related' enzyme that belongs to the cupin superfamily, which uses a non-heme ferrous iron ($Fe^{2+}$) as a cofactor and facilitates the integration of molecular oxygen into a myriad of biomolecules. Members of this superfamily feature a highly conserved $Fe^{2+}$-binding motif consisting of two histidines and an acidic residue (Glu/Asp), known as the 'facial triad'[8–10]. In plants, the 2OG oxygenases are involved in the biosynthesis of important molecules such as anthocyanin, gibberellin, auxin and ethylene[11]. ACO uses ascorbate, rather than 2-oxoglutarate as the co-substrate, and requires $CO_2$, one of the products of the ACO reaction, as an activator[12,13].

The crystal structure of Petunia hybrida ACO has been determined by Schofield et al.[14]. In combination with biochemical and molecular docking studies, a possible catalytic mechanism of ACO has been proposed[15–18]. During catalysis, ACC binds to $Fe^{2+}$ in a bidentate manner via both its carboxylate and amino groups, together with the facial triad. This special arrangement allows $O_2$ binding to $Fe^{2+}$ to complete the octahedral coordination. In the presence of bicarbonate, ascorbate reduces the $Fe=O_2$ and breaks the O–O bond to form a high-valent Fe intermediate (likely a Fe(IV)–oxo species), which promotes ACC-ring opening to generate ethylene. The reactive Fe is then reduced by ascorbate to regenerate the catalytically active $Fe^{2+}$.

Identification of new analogues, agonists and inhibitors of plant hormone pathways and their application in basic research and agricultural practice has been well documented[19–21]. Due to the important function of ethylene in controlling processes such as fruit ripening and flower senescence, ethylene production and perception are important pathways for industrial intervention. For example, 1-methylcyclopropene, a competitive inhibitor of ethylene perception, has been extensively used to maintain freshness of ornamental plants and fruits under several brand names[22,23]. Aminoethoxyvinylglycine (AVG) has been used to inhibit ethylene biosynthesis as an inhibitor of ACC synthase[24].

However, AVG is non-specific to ethylene biosynthesis as it likely affects all or most pyridoxal-5′-phosphate-dependent enzymes, such as the Trp aminotransferase in indole-3-acetic acid biosynthesis[25,26]. Aminooxyacetic acid was also reported to inhibit the activity of ACC synthase by forming a complex with the cofactor pyridoxal-5′-phosphate[27]. Analogues of ACC, such as α-aminoisobutyric acid (AIB) and 2-aminooxyisobutyric acid (AOIB), were documented to inhibit ethylene formation by competitively targeting ACO, but with a very low inhibition efficacy[28,29]. Cobalt ion was also reported to inhibit ethylene synthesis by interfering with $Fe^{2+}$ that is required for ACOs and other 2OG catalysis[30]. Due to the low efficacy of AIB/AOIB and non-specific effect of AVG, Aminooxyacetic acid or cobalt ion, there is to date no effective means to specifically manipulate ethylene production at the point of ACS and ACO regulation.

Here, we have applied a phenotype-based chemical biology approach and identified pyrazinamide (PZA) as an ethylene biosynthesis inhibitor. PZA is well-documented in clinics as an important drug to treat tuberculosis[31,32]. The primary action of PZA is to function as a pro-drug, and is converted to pyrazinecarboxylic acid (POA) by the mycobacterial enzyme pyrazinamidase (PZase)/nicotinamidase[33]. POA is proposed to execute its anti-tuberculosis effect by causing cytoplasmic acidification and de-energize the membrane[34], binding to bacterial ribosomal protein RpsA to inhibit trans-translation[35], and inhibiting bacterial aspartate decarboxylase PanD[36]. We demonstrate that PZA is also converted to POA in plants, which directly binds to the ACO proteins and inhibits their enzyme activity. We further determine the crystal structures of Arabidopsis ACO2 in complex with POA or 2-Picolinic Acid (2-PA, a POA-related compound), which reveal interaction details between ACO and POA/2-PA. Our studies demonstrate a different function of PZA in the control of plant ethylene biosynthesis, which holds enormous implication and potential in agriculture and horticulture.

## Results

**Identification of PZA as an ethylene biosynthesis inhibitor.** To isolate inhibitors of ethylene biosynthesis and signalling, we screened a chemical library (SP 2000, http://www.msdiscovery.com) for suppressors of the constitutive ethylene response phenotype observed in the mutants of eto1-2 (ethylene overproducer1-2) and ctr1-1 (constitutive triple response1-1). eto1-2 is an ethylene biosynthesis mutant that shows enhanced ACS protein stability[37], whereas ctr1-1 is a signalling mutant that shows constitutively activated ethylene response[38]. The 3-day-old etiolated seedlings of eto1-2 and ctr1-1 exhibited a typical 'triple response' phenotype, including thickening and shortening of hypocotyl and root, as well as pronounced apical hook (Fig. 1a). Approximately 10 seeds were germinated for 3 days in dark in each well of 96-well microplates containing one chemical in the library at a concentration of $5 \mu g \, ml^{-1}$. After two rounds of screening, one of the chemicals, PZA, was identified as a potent ethylene biosynthesis inhibitor, as it partially suppressed the short-root and short-hypocotyl phenotype of eto1-2, but had little effect on several ethylene signalling mutants, including the constitutive ethylene response mutant ctr1-1 and ethylene-insensitive mutant ein2-5 (Fig. 1a–c). Interestingly, treatments of wild-type (Col-0) etiolated seedlings with increasing doses of PZA slightly promoted the lengths of their hypocotyls and roots (Fig. 1b,c), suggesting that PZA can also suppress the basal level of ethylene production. We further found that PZA application promoted cell elongation in the root maturation zone of eto1-2, but had little effect on the size of the cell division zone (Supplementary Fig. 1a), consistent with the finding that ethylene inhibits root elongation mainly through suppressing the cell

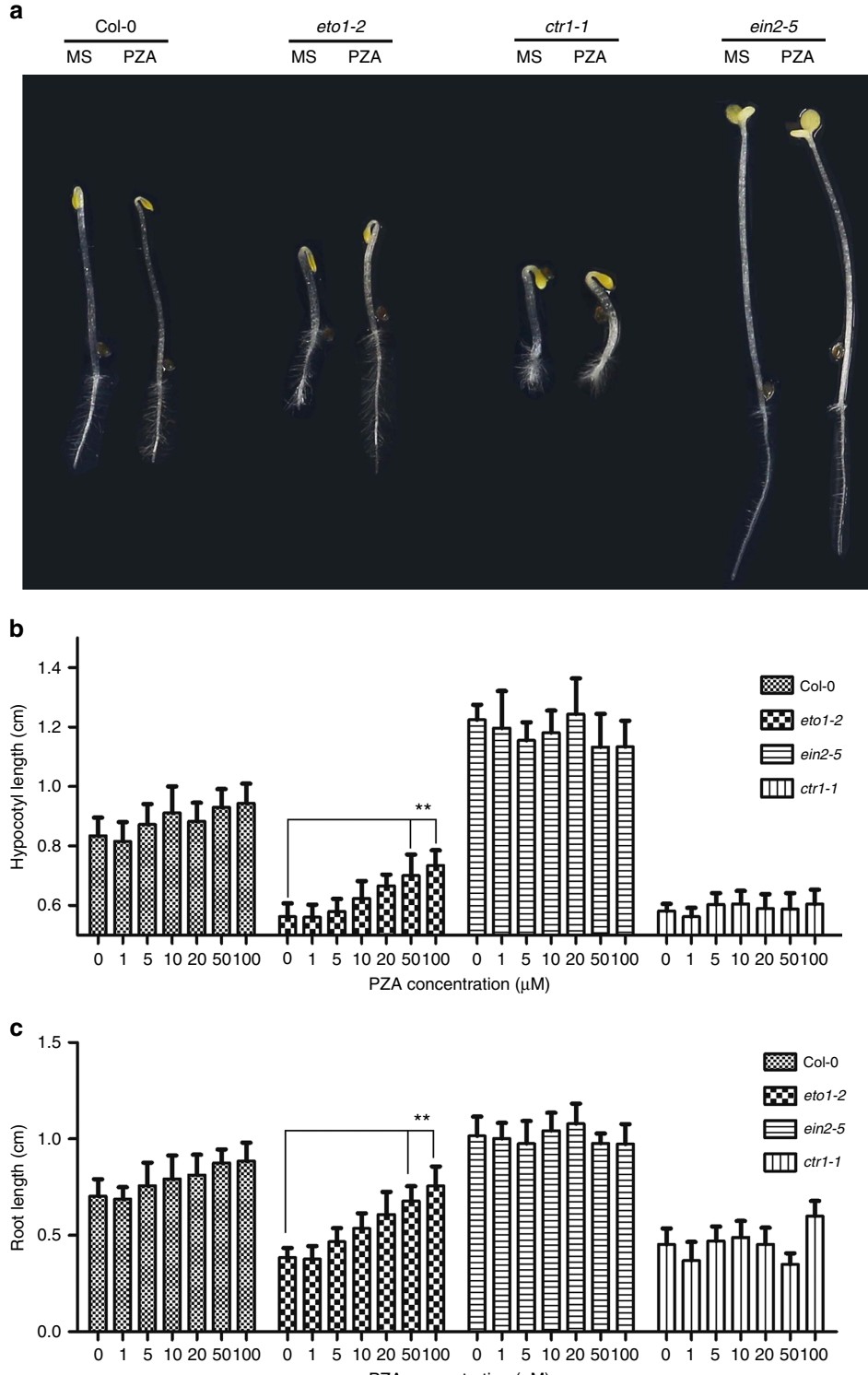

**Figure 1 | PZA specifically suppresses the short-root and short-hypocotyl phenotype of *eto1-2*.** (**a**) 3-day-old etiolated seedlings of Col-0, *eto1*-2, *ctr1-1* and *ein2-5* grown on horizontally oriented MS only or MS medium supplemented with 50 μM PZA. Quantification of the hypocotyl lengths (**b**) and root lengths (**c**) of 3-day-old etiolated seedlings of Col-0, *eto1*-2, *ctr1-1* and *ein2-5* grown on MS only or MS medium supplemented with different concentrations of PZA (1, 5, 10, 20, 50 and 100 μM). Bars represent the average length ( ± s.d.) of fifteen seedlings (Student's *t*-test, between PZA-treated and non-treated seedlings; **$P < 0.01$).

elongation of root maturation zone[39]. Moreover, PZA treatment also effectively suppressed the root hair formation of *eto1-2* (Supplementary Fig. 1b). Therefore, PZA specifically suppresses growth inhibition and root hair development of the ethylene overproducing mutant *eto1-2*.

**PZA inhibits the last step of ethylene biosynthesis**. To further determine how PZA influences ethylene biosynthesis, we examined the effect of PZA on ethylene- and ACC-induced ethylene responses. PZA treatment significantly suppressed the ACC-induced short hypocotyl and root phenotype, but had no

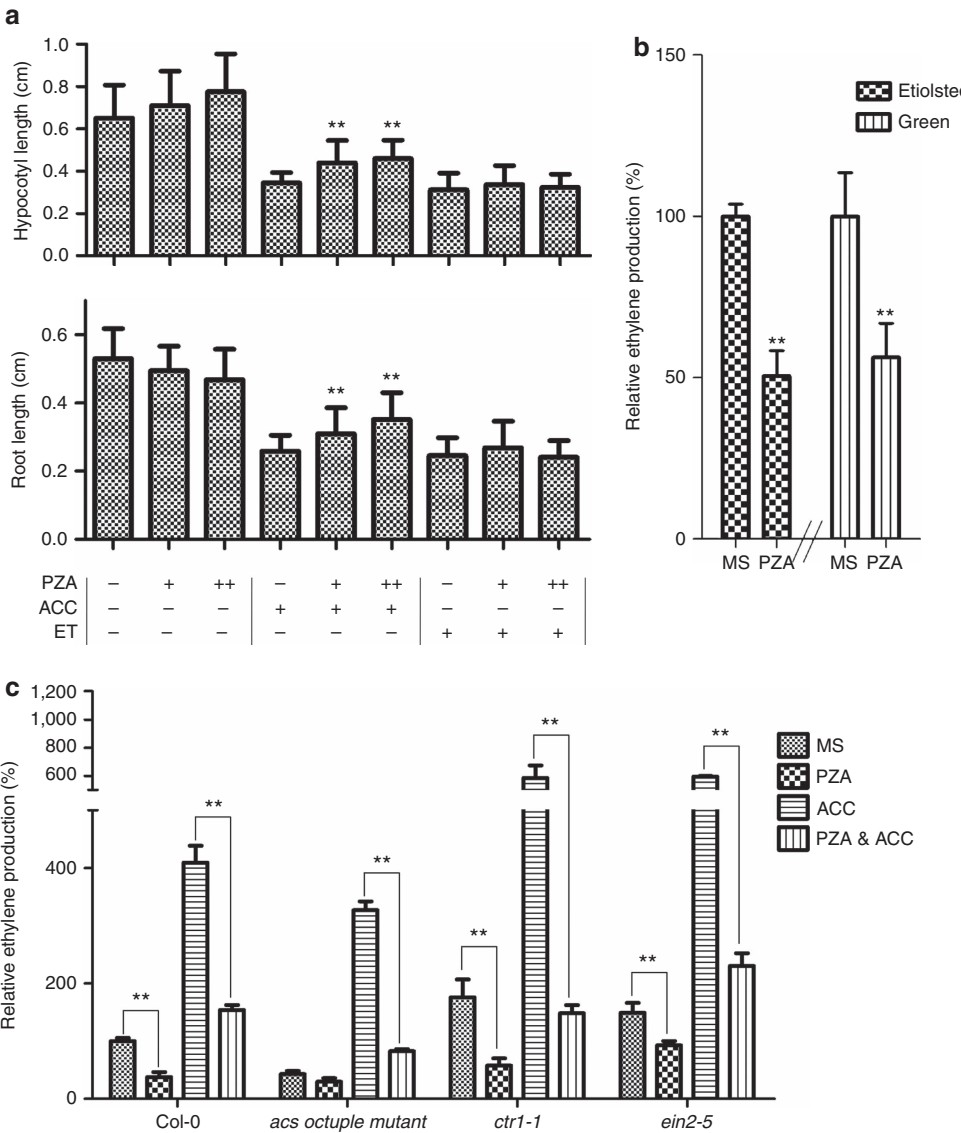

**Figure 2 | PZA inhibits the last step of ethylene biosynthesis.** (**a**) Quantification of the hypocotyl lengths (upper) and root lengths (lower) of 3-day-old etiolated seedlings of Col-0 grown on vertically oriented MS medium supplemented with 1 μM ACC or 2 p.p.m. ethylene (ET) and/or 50 μM ( + ) or 100 μM ( + + ) PZA. Bars represent the average length ( ± s.d.) of fifteen seedlings (Student's *t*-test, between PZA-treated and non-PZA-treated seedlings; \*\**P*<0.01). (**b**) Quantification of the relative ethylene production of 30 3-day-old etiolated seedlings and 5-day-old green seedlings of *eto1-2* with 50 μM PZA compared to MS treatment. Bars represent the means ( ± s.d.) of five independent treatments (Student's *t*-test, between PZA-treated and non-treated seedlings; \*\**P*<0.01). (**c**) Quantification the relative ethylene production (compared to Col-0 under MS treatment) of 3-day-old etiolated seedlings of Col-0, *acs* octuple mutant, *ctr1-1* and *ein2-5* with the application of 10 μM ACC and/or 50 μM PZA for 24 h. Bars represent the means ( ± s.d.) of three independent treatments (Student's *t*-test, between PZA-treated and non-treated seedlings; \*\**P*<0.01).

effect on ethylene-induced phenotype (Fig. 2a). This result indicates that PZA functions upstream of ethylene perception and downstream of ACC formation, implying that PZA could modulate the activity of ACO or ACC conjugating enzymes to influence ethylene production[3,40].

We then investigated whether PZA application decreases ethylene production of *eto1-2*. Our results showed that PZA effectively inhibited the ethylene production in both etiolated and green seedlings of *eto1-2* (Fig. 2b). As for *eto1-2*, PZA showed similar inhibition pattern of ACC-induced ethylene production in wild type, *acs* octuple mutant (defective on eight *ACS* genes), *ctr1-1* and *ein2-5*, implying that PZA inhibition of ethylene biosynthesis is not dependent on ACS genes and the ethylene signalling pathway (Fig. 2c). PZA application also

inhibited the ACC-induced EIN3 protein accumulation and transcriptional activation of target promoter[41,42] (Supplementary Fig. 2a,b). In addition, to determine whether PZA affects the activity of ACC conjugating enzymes, whose action could also decrease the pool of free ACC in the cells[40], we measured the content of ACC and the major ACC conjugate form, 1-(malonylamino) cyclopropane-1-carboxylic acid (M-ACC)[43] in *eto1-2* under PZA treatment. Our results showed that PZA application led to a slight increase on ACC level and almost no change on M-ACC level, suggesting that PZA had little effect on the ACC conjugating enzymes (Supplementary Fig. 3). On the basis of these results, we conclude that PZA suppresses the ethylene response by inhibiting ACO that converts ACC to ethylene.

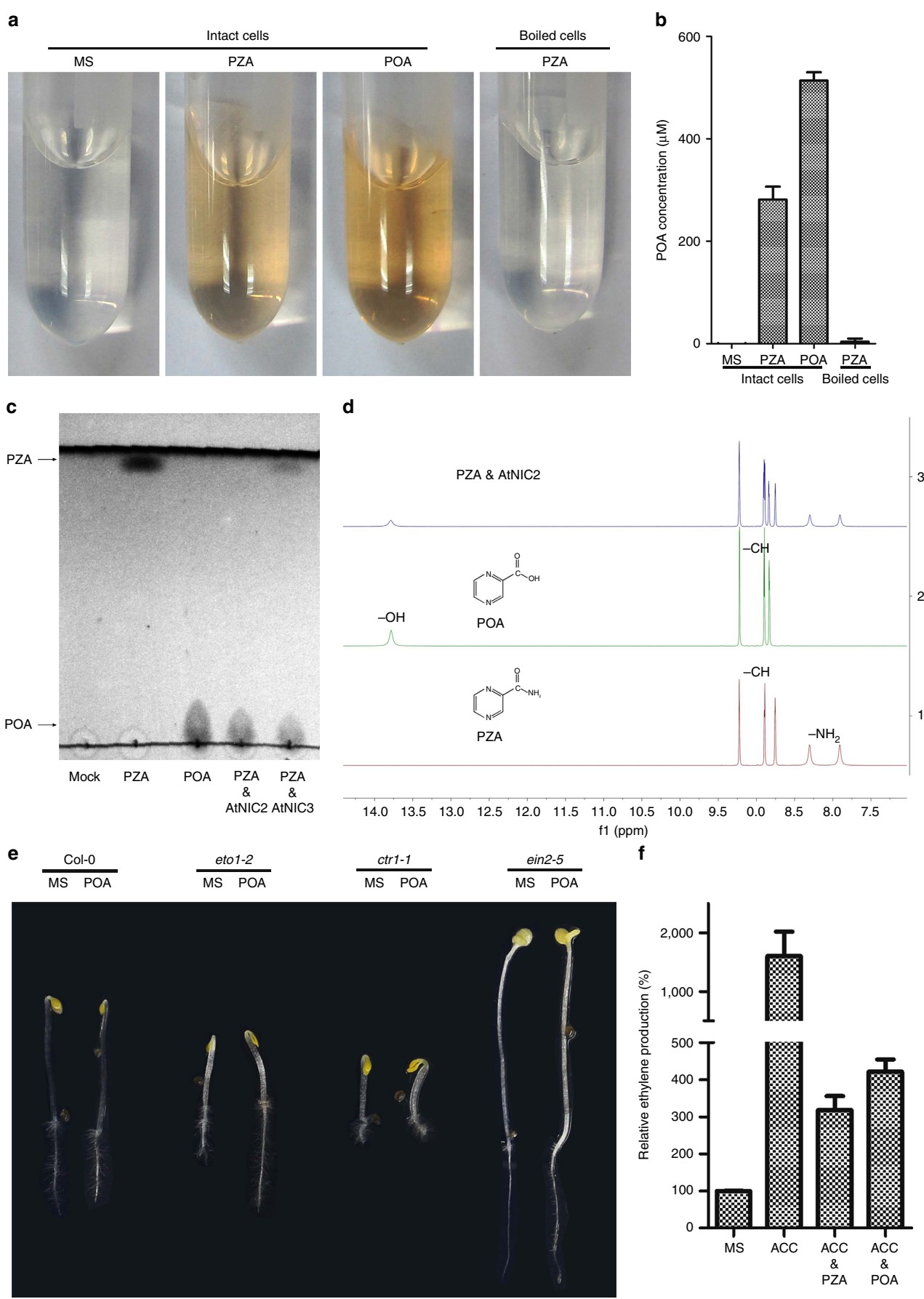

**POA is the active form of PZA**. Next, we tested whether PZA directly affects the enzymatic activity of ACO. There are five ACO proteins (AtACO1-5) in the *Arabidopsis* genome, which show high degrees of sequence identity to each other. We purified recombinant AtACO2 protein expressed in *Escherichia coli*, and performed *in vitro* ACO enzyme assays (Supplementary Fig. 4a). We measured ACO activity by quantifying the production of ethylene using GC-FID (Gas Chromatography Flame Ionization Detector)[44,45]. Interestingly, PZA showed no inhibition to the activity of AtACO2, even at a high concentration (2 mM) (Supplementary Fig. 4b).

PZA is a pro-drug in the treatment of tuberculosis infection[31], which is converted to POA to execute its anti-tuberculosis effect[42]. These results prompted us to test whether POA could be the active form of PZA to inhibit ACO activity. PZA treatment inhibited ethylene production in the *Arabidopsis* PSB-D suspension cultured cells as in the seedlings (Supplementary Fig. 5). By employing the Wayne Test that is used to quantitatively measure POA[46,47], we found that PZA was indeed converted into POA in the PSB-D culture (Fig. 3a). We further confirmed the presence of POA in the PZA-treated culture cells using mass spectrometry (Supplementary Fig. 6). Notably, POA formation was abolished when the cultured cells were killed first by boiling, suggesting that such conversion results from an enzymatic activity rather than a spontaneous process (Fig. 3b).

Three homologues of PZase/nicotinamidase are present in *Arabidopsis*: AT2G22570, AT5G23220 and AT5G23230, which are also known as AtNIC1, AtNIC2 and AtNIC3, respectively. These enzymes catalyse the formation of nicotinic acid (NA) from nicotinamide and play essential roles in NAD$^+$ biosynthesis[33]. While no combinatory mutant harbouring all three *AtNIC* gene mutations was available, single mutant of AtNIC2 (Stock: CS411587) showed no obvious effect of ethylene production upon PZA inhibition (Supplementary Fig. 7), suggesting that these enzymes could be functionally redundant. To directly test whether *Arabidopsis* NIC proteins are functional PZases, we purified GST-tagged AtNIC2 and AtNIC3 from *E. coli* and performed *in vitro* PZase reactions. The production of POA was examined using thin layer chromatography (TLC) and visualized by ultraviolet light. Both AtNIC2 and AtNIC3 were able to convert PZA into POA (Fig. 3c). The identity of the resulting POA was further confirmed by nuclear magnetic resonance (NMR) analysis (Fig. 3d). Collectively, these results demonstrate that PZA can be converted into POA by the *Arabidopsis* nicotinamidases.

The influence of POA application on ethylene response and ethylene biosynthesis of *Arabidopsis* seedlings was then investigated. Indeed, POA specifically suppressed the short-root and short-hypocotyl phenotype of *eto1-2* etiolated seedlings (Fig. 3e) and inhibited the ACC-induced ethylene production in a manner similar to PZA (Fig. 3f). Taken together, these results suggest that

POA is very likely the active form of PZA in plants to inhibit ethylene biosynthesis.

**POA selectively inhibits enzymatic activity of ACO**. Next, we tested whether POA directly inhibits the activity of ACO. In contrast to PZA, POA inhibits the activity of AtACO2 in a concentration dependent manner (IC$_{50}$ = 72.05 ± 1.11 μM) (Fig. 4a). POA did not obviously change the stability of purified AtACO2 protein (Supplementary Fig. 8), supporting its role on ACO enzymatic activity. We then performed enzyme kinetic analyses, and the double-reciprocal plots (Lineweaver–Burk plots) suggested that POA is likely a competitive inhibitor of ACO ($K_i$ = 89.84 ± 18.32 μM; error represents s.d.) (Fig. 4b). Likewise, the other four ACO proteins in *Arabidopsis* were also inhibited by POA with $K_i$ values ranging from 186.8 to 721.7 μM (Fig. 4c).

To determine the binding affinity between POA and AtACO2, we performed isothermal titration calorimetry (ITC) experiments. POA interacted with AtACO2 in the present of Fe$^{2+}$, with a dissociation constant ($K_D$) of 14.8 μM (Fig. 4d). To evaluate the binding specificity of POA to ACO, we chose three representative members of 2OG-oxygenase superfamily in *Arabidopsis*: gibberellin C2 oxidase 2 (GA2OX2), anthocyanidin synthase (ANS) and dioxygenase for auxin oxidation (DAO), which have important roles in gibberellin biosynthesis, anthocyanin production and auxin metabolism, respectively[11,48–50]. By ITC measurements, we detected no interaction between POA and these three proteins (Fig. 4d). Accordingly, we also monitored the GA response (using *Arabidopsis* seedlings harbouring a *pRGA::GFP-RGA* reporter[51]), anthocyanin content and auxin response (using *pDR5::GFP* reporter lines[52]) upon PZA treatment. PZA treatment showed virtually no effect on GA response and anthocyanin accumulation, and affected auxin response in an EIN2-dependent manner (Supplementary Fig. 9)[53,54], indicating that POA did not influence all three pathways directly. Taken together, these results suggest that PZA/POA is not a wide-spectrum 2OG-oxygenase inhibitor, but rather selectively targets the ACO enzymes to inhibit ethylene production.

**SAR analysis of POA derivatives**. To gain insight into the underlying structure-activity relationship (SAR) of POA, we obtained several POA derivatives and measured their effects on the activity of ACO (Fig. 5a). Among the compounds tested, only 2-picolinic acid (2-PA) directly inhibited the enzymatic activity of AtACO2 like POA (Fig. 5b). Notably, the inhibition effect of 2-PA was stronger than that of POA, while a known ACO inhibitor, AIB, showed much weaker inhibition effect than POA in our assay (Fig. 5b). Consistently, 2-PA and its corresponding amide, 2-picolinamide, remarkably inhibited ethylene production in *Arabidopsis* seedlings and suspension cultured cells (Supplementary Fig. 10a–c). Together with the finding that PZA failed to inhibit AtACO2 activity directly

**Figure 3 | POA is the active form of PZA in *Arabidopsis*.** (**a**). The Wayne Test result of PZA application in suspension cultured cells. Cells were cultured in liquid MS or MS medium supplemented with 500 μM PZA or POA for 24 h. The boiled cells with PZA worked as a negative control. (**b**) Quantification of the POA concentration shown in **a**. POA concentrations are calculated based on the corresponding 450 nm absorption and the POA-Abs$_{450}$ standard curve. Bars represent the average concentration ( ± s.d.) of three independent treatments. (**c**) TLC (thin layer chromatography) analysis demonstrating that PZA can be hydrolyzed by *Arabidopsis* NIC2 and NIC3 *in vitro*. Reaction products were separated by silica TLC plates developing using dichloromethane:methanol (3:1) and detected with ultraviolet. (**d**) NMR analysis of reaction products of PZA catalyzed by AtNIC2. Products were extracted using ethyl acetate and analysed using Bruker-400 MHz NMR in Dimethyl sulfoxide-d6. The middle panel refers to the POA standard sample whereas the bottom panel is for PZA. The peaks corresponding to the carboxylic acid proton of POA (-OH) and the two amino protons of PZA (-NH$_2$) are denoted. The upper panel indicates the reaction mixture after PZA was treated with AtNIC2, wherein the carboxylic acid proton peak was observed, indicative of the production of POA. (**e**) 3-day-old etiolated seedlings of Col-0, *eto1-2*, *ctr1-1* and *ein2-5* grown on horizontally oriented MS only or MS medium supplemented with 50 μM POA. (**f**) Quantification of the relative ethylene production (compared to MS treatment) of 3-day-old etiolated seedlings of Col-0 with the application of 100 μM ACC and/or PZA or POA (500 μM) for 24 h. Bars represent the means ( ± s.d.) of five independent treatments.

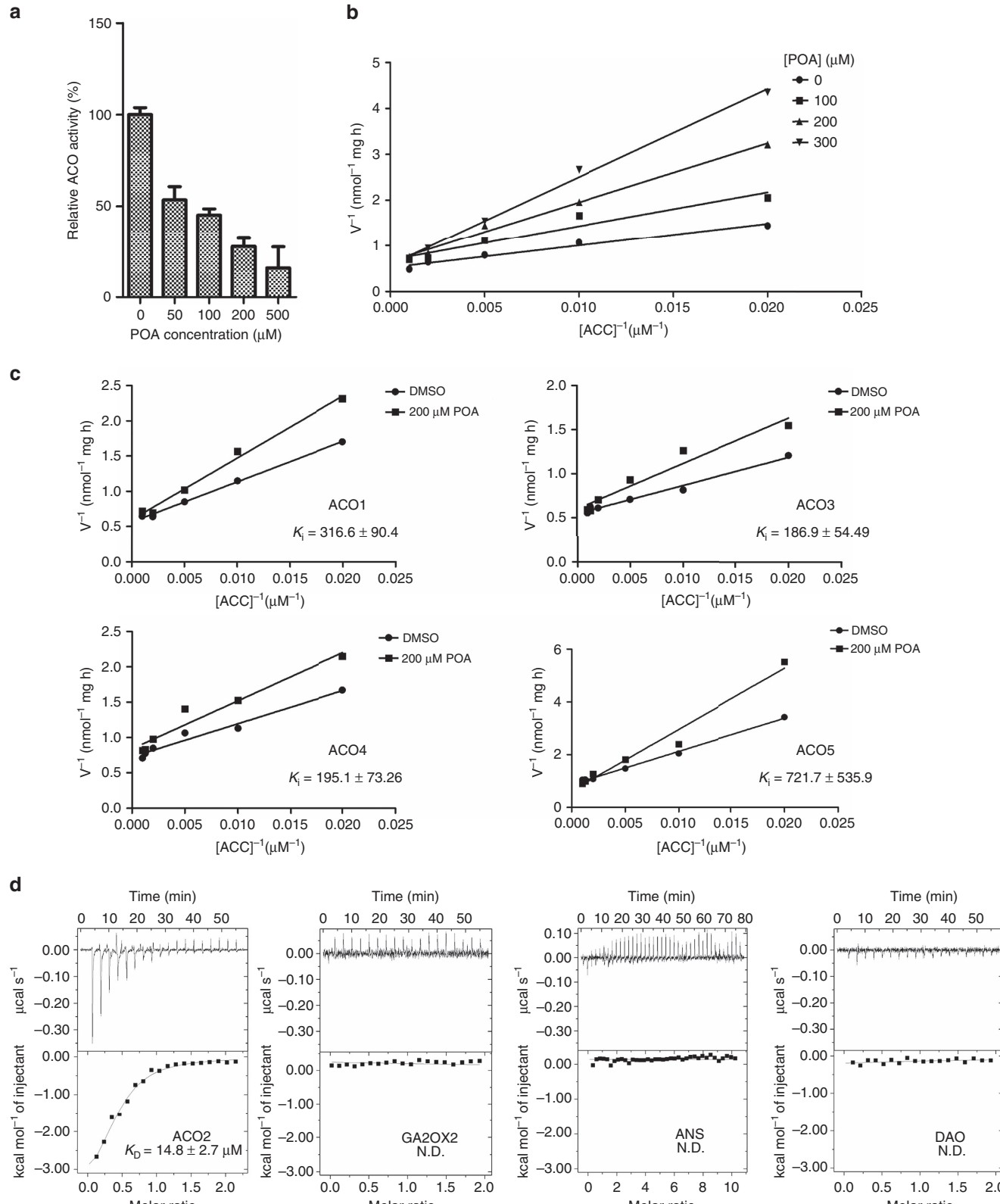

**Figure 4 | POA selectively inhibits the enzymatic activity of *Arabidopsis* ACO enzymes.** (**a**) Quantifications of the relative AtACO2 activity treated with different concentrations of POA. Values represent the mean (±s.d.) of three replicates, and the experiment was carried out twice with similar results. (**b**) Kinetic analysis of AtACO2 inhibition by POA. Activity assays were performed with varying concentration of ACC in the absence and presence of 100, 200 or 300 μM POA. Double-reciprocal plots of initial velocities (Lineweaver–Burk plots) showing a competitive inhibition. The experiment was carried out twice with similar results. (**c**) Kinetic analysis of the inhibition by POA to AtACO1, AtACO3, AtACO4 and AtACO5. Activity assays were performed with varying concentration of ACC in the absence (+DMSO) and presence of 200 μM POA. The experiment was carried out twice with similar results. (**d**) The interactions between POA and AtACO2, GA2OX2, ANS as well as DAO were measured using ITC. N.D., not determined.

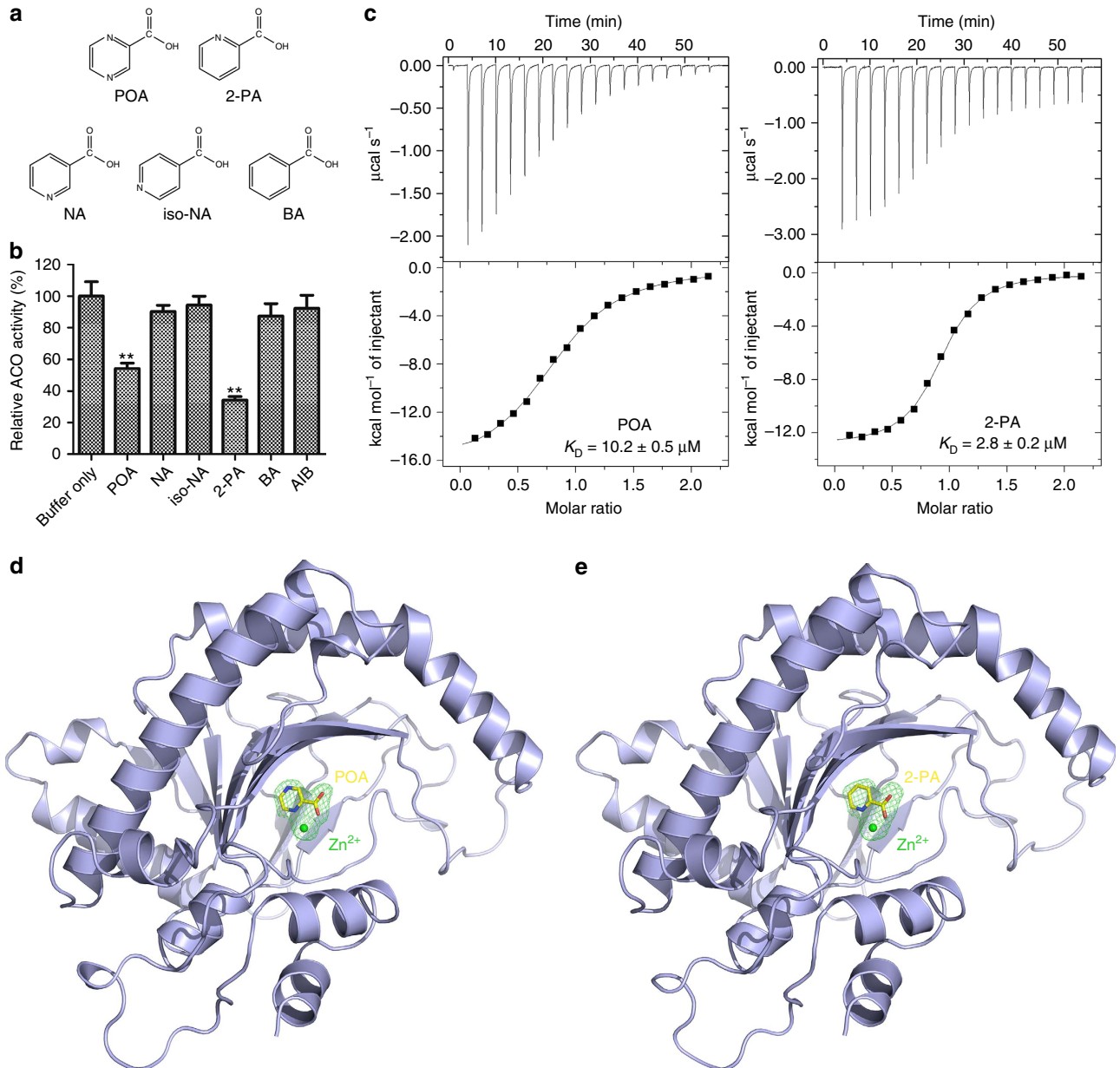

**Figure 5 | Crystal structures of AtACO2–POA and AtACO2–2-PA complexes.** (**a**) Chemical structures of POA derivatives. (**b**) Quantifications of the relative AtACO2 activity treated with different POA derivatives (200 μM). Values represent the mean ( ± s.d.) of three replicates, and the experiment was carried out twice with similar results (Student's *t*-test, between PZA-treated and non-PZA-treated seedlings; **$P < 0.01$). (**c**) POA (left) or 2-PA (right) binds to AtACO2 in the presence of $Zn^{2+}$ as measured by ITC, with dissociation constant ($K_D$) as indicated. (**d**) The crystal structure of AtACO2–POA complex. AtACO2 is shown as ribbons and coloured in light blue. The Fo–Fc electron density map (contoured at $3\sigma$) which reveals the presence of POA and $Zn^{2+}$ is shown as a green mesh. The POA and $Zn^{2+}$ are omitted to calculate the map. (**e**) The crystal structure of ACO2–2-PA complex, depicted as in (**c**). BA, benzoic acid; NA, nicotinic acid; Iso-NA, iso-nicotinic acid; 2-PA, 2-picolinic acid.

(Supplementary Fig. 4b), these results suggest that the carboxylic group and the nitrogen in its ortho-position are important for the inhibitory function of POA.

**Structures of ACO-POA and ACO-2-PA complexes.** To elucidate how POA or 2-PA interacts with ACO, we sought to obtain the crystal structure of ACO in complex with these compounds. Since $Fe^{2+}$ is highly unstable and rapidly converted to $Fe^{3+}$ in aerobic conditions, in order to facilitate crystallization, stable divalent metal ions are often used as surrogates to study the 2OG proteins, especially their interaction with inhibitors[8,55].

We found that $Zn^{2+}$ can effectively mediate the interaction between AtACO2 and POA or 2-PA (Fig. 5c). In the presence of $Zn^{2+}$, POA and 2-PA bound to AtACO2 with dissociation constants ($K_D$) of 10.2 and 2.8 μM, respectively. $Zn^{2+}$ did not support the catalytic activity of ACO (Supplementary Fig. 11), similar to its effect on other 2OG oxygenases[56].

We subsequently co-crystallized AtACO2 with POA or 2-PA in the presence of $Zn^{2+}$, and determined both structures at 2.1 Å (Table 1; Fig. 5d,e and Supplementary Fig. 12). Overall, the structure of AtACO2 resembles the *Petunia hybrid* ACO (PhACO), featuring a double-stranded-helix jellyroll fold surrounded by α-helices. When the two structures are

**Table 1 | Crystal data collection and refinement statistics.**

|  | ACO2/POA | ACO2/2-PA |
|---|---|---|
| *Data collection* |  |  |
| Space group | P2$_1$ | P1 |
| Cell dimensions |  |  |
| *a, b, c* (Å) | 49.38, 95.10, 70.15 | 69.46, 95.34, 95.89 |
| α, β, γ (°) | 90, 105.91, 90 | 90.07, 89.48, 89.99 |
| Resolution (Å) | 50–2.1 (2.14–2.1) | 50–2.1 (2.14–2.1) |
| $R_{merge}$ | 0.07 (0.16) | 0.11 (0.43) |
| $I/\sigma I$ | 25.0 (13.2) | 10.0 (2.3) |
| Completeness (%) | 99.0 (98.3) | 97.4 (96.8) |
| Redundancy | 3.6 (3.6) | 3.5 (3.5) |
|  |  |  |
| *Refinement* |  |  |
| Resolution (Å) | 2.1 | 2.1 |
| No. reflections | 35,734 | 140,503 |
| $R_{work}/R_{free}$ | 15.3/20.9 | 17.1/21.8 |
| No. atoms |  |  |
| Protein | 4,756 | 18,880 |
| Ligand/ion | 18/2 | 72/8 |
| Water | 420 | 1,274 |
| *B*-factors |  |  |
| Protein | 31.6 | 38.1 |
| Ligand/ion | 25.6 | 23.1 |
| Water | 36.2 | 37.5 |
| R.m.s. deviations |  |  |
| Bond lengths (Å) | 0.008 | 0.008 |
| Bond angles (°) | 0.860 | 1.061 |

Each dataset was collected from a single crystal. Values in parentheses are for highest-resolution shells.

superposed, the root-mean-square deviation is 1.2 Å over 242 aligned Cα atoms. The main differences are located in regions that are involved in forming the tetramer structure of PhACO. In AtACO2, the α-helix corresponding to PhACO α3 has a curved conformation; and the α-helix corresponding to PhACO α11 folds back rather than extends to interact with another molecule. Both changes would lead to the disruption of the protein interfaces seen in the PhACO tetramer (Supplementary Fig. 13). Consistently, AtACO2 is a monomer in solution as suggested by gel filtration analyses (Supplementary Fig. 14).

POA and 2-PA bind to AtACO2 in a similar manner (Fig. 6a,b). In both structures, the $Zn^{2+}$ ion is coordinated by the facial triad, including His180, Asp182 and His237, and binds at a similar position as $Fe^{2+}$ in PhACO. POA or 2-PA in turn binds to the $Zn^{2+}$ via the carboxylate group and the N-2 nitrogen in the pyrazine or pyridine ring. Mutating His180 to alanine abolished the interaction between AtACO2 with POA or 2-PA (Fig. 6c,d). Although no structure has been determined for an ACO protein in complex with ACC, it is proposed that ACC also binds directly to the $Fe^{2+}$ via its carboxylate and amino groups during catalysis, and mutation in the facial triad led to inactivating the AtACO2 protein activity (Supplementary Fig. 15). As such, POA or 2-PA would exclude ACC from coordinating with the metal ion, consistent with our kinetic analyses showing that POA is a competitive inhibitor of ACO (Fig. 4b).

The carboxylate group of POA or 2-PA forms a polar interaction with the side chain of Lys161. Lys161 is hypothesized to interact with a bicarbonate ion, which is required to activate the enzyme during catalysis[18]. Mutating Lys161 to alanine reduced the binding between ACO2 and POA or 2-PA and displayed impaired catalytic activity (Fig. 6c,d and Supplementary Fig. 15). Lys291 contributes to the hydrogen bond interactions with POA or 2-PA through a water molecule. Besides these polar interactions, the pyrazine/pyridine ring structure also forms

hydrophobic/van der Waals interactions with residues including Ile187, Leu189, Ala251 and Phe253. When Ala251 or Phe253 was mutated together with Lys161, the two double mutants (K161A/A251L, K161A/F253A) display completely abolished binding to POA or 2-PA and no detectable activity (Fig. 6c,d and Supplementary Fig. 15).

Collectively, these results provide a molecular explanation of how POA and 2-PA bind to AtACO2 and inhibit its enzyme activity. Notably, the residues that are involved in binding to these molecules are highly conserved in the ACO family of proteins (Supplementary Fig. 16), suggesting that POA and 2-PA could function as broad-spectrum ethylene biosynthesis inhibitors in plants.

## Discussion

Using a phenotype-based screen of a selected chemical library containing 2,000 diverse generic drugs, we have identified a small compound (PZA) that specifically suppresses the ethylene responses of *eto1-2* and inhibits ethylene biosynthesis in *Arabidopsis*. Further studies revealed that PZA can be converted to POA in *Arabidopsis* cells, which is the active form of PZA to inhibit ethylene biosynthesis. The combination of genetic, biochemical and structural biology approaches unequivocally demonstrate that POA directly targets and inhibits ACO to suppress the final step of ethylene biosynthesis. The SAR assays revealed the importance of its carboxylic group and the nitrogen in the ortho-position of POA for ACO binding and inhibition, which was validated by the crystal structure of the AtACO2–POA complex. Taken together, our data suggest a model for POA inhibition of ethylene biosynthesis, in which POA specifically binds to ACO and competes with its substrate ACC for the coordination with ferrous iron in the catalytic centre. Although a number of small molecules have been identified in plant chemical biology studies[57], the target identification of these

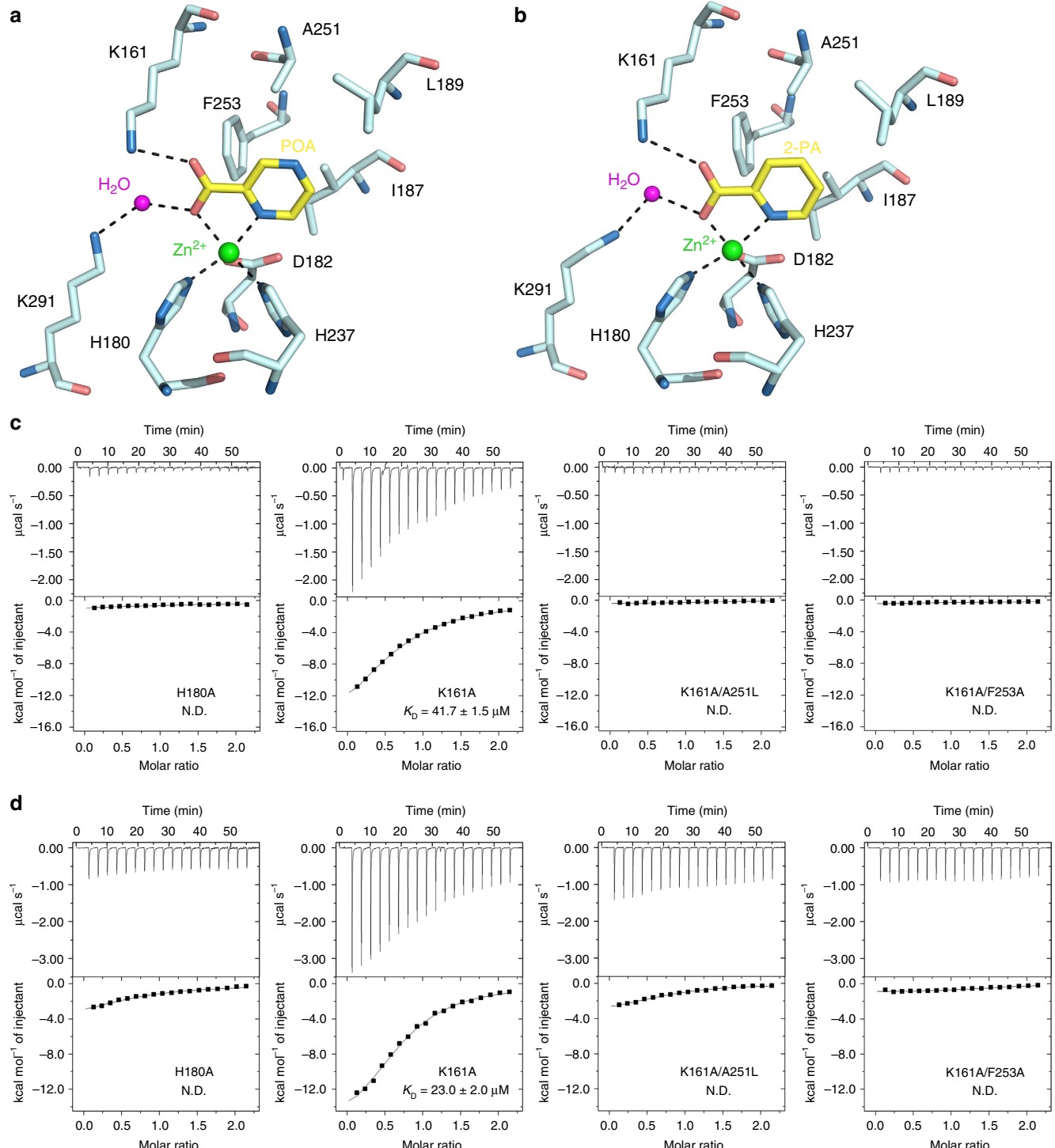

**Figure 6 | Critical residues of AtACO2 involved in binding to POA and 2-PA.** (**a**) Detailed molecular interactions between POA and AtACO2. AtACO2 residues involved in POA binding are shown and labelled. POA is shown as sticks, and its carbon atoms are coloured in yellow. The $Zn^{2+}$ and water molecule are shown as green and magenta spheres, respectively. Salt bridge and hydrogen bond interactions are shown as dashed lines. (**b**) Molecular interaction between 2-PA and ACO2, depicted as in **a**. (**c**) The binding between POA and the AtACO2 mutants were measured using ITC in the presence of $Zn^{2+}$. All mutants display impaired or abolished binding to POA. (**d**) AtACO2 mutants also display impaired or abolished binding to 2-PA. N.D., not determined.

chemicals still remains a big challenge in most cases. Based on this study, PZA/POA belong to the few inhibitors whose target proteins and the mode of action have been well elucidated.

An efficient and specific inhibitor of ethylene biosynthesis not only has great prospect in the basic research of ethylene biology, but also benefits postharvest food storage and management in

agriculture and horticulture. Although ACS is generally considered as the rate-limiting step in ethylene biosynthesis, there is growing evidence that ACO acts as a regulatory point in ethylene production, as ACO gene family members show differential regulation in response to various developmental and environmental cues[5–7]. The application of POA will not only

overcome the gene redundancy of ACO gene family, but also benefit the study on the fast regulation of ethylene production in response to myriad stimuli. Compared with many known ACO inhibitors, such as cobalt ion, AIB or AOIB, POA displays dramatically improved efficacy and specificity. The SAR analyses identified 2-PA as an even more efficient inhibitor than POA. As such, there appears to be considerable room to further increase the potency of PZA/POA by modifying their chemical structures. The crystal structure of AtACO2 in complex with POA or 2-PA would no doubt serve as a foundation to design and optimize more desirable compounds to control plant ethylene production.

PZA is an important front-line prodrug that helps to shorten the therapy of tuberculosis from previously 9–12 months to 6 months[31,58]. Despite its importance for tuberculosis treatment, the mode of PZA action as a bactericidal agent is still unclear. Once entering the bacillus, PZA is converted to POA by the mycobacterial PZase/nicotinamidase, a process also observed in *Arabidopsis* by our study. The sterilizing effect of POA is proposed to have several aspects. First, POA can cause cytoplasmic acidification and disruption of membrane energy and function through export of POA$^-$ and re-import of protonated POA[34]; Second, POA directly binds to multiple targets and inhibits their functions. For instance, two potential targets of POA in *M. tuberculosis* have been recently reported: the ribosomal protein S1 (RpsA) and the aspartate decarboxylase (PanD)[35,36]. However, increasing body of evidence suggests the existence of additional PZA/POA-binding targets[32]. Our study provides information about the targets of POA and the mode of its action. For example, POA requires ferrous ion to bind to ACO, a subfamily member of 2OG oxygenases. Interestingly, it was known that the anti-tuberculosis activity of PZA or POA is significantly enhanced by iron or under hypoxic/anaerobic conditions[59]. Moreover, several homologues of 2OG oxygenases are present in the genome of *M. tuberculosis*[60]. It is thus compelling to investigate whether some of these proteins are also targeted and inhibited by POA. The crystal structure of the C-terminal domain of RpsA in complex with POA was recently determined[61]. The molecular details revealed by our structure data and mutations analysis of ACO in complex with POA provide a quite different picture on the POA-binding pocket and its inhibitory mechanism. In this regard, our findings in *Arabidopsis* may help identify other targets of these molecules and uncover distinctive mechanism of PZA/POA in human tuberculosis therapy.

## Methods

**Plant material and growth conditions.** All plant material employed in this study are *Arabidopsis* Col-0 wild-type plants or transgenic lines and mutants in the Col-0 background. *eto1-2* (ref. 37), *ctr1-1* (ref. 38), *ein2-5*, *acs* octuplet mutant[62], pRGA::GFP-RGA[51], pDR5::GFP/Col-0 (ref. 52) lines were described previously. pDR5::GFP/*ein2-5* was generated by genetic crosses. CS411587 (NIC2 single T-DNA insertion mutant) was purchased from Arabidopsis Biological Resource Center. Surface-sterilized seeds were sown in 96-well microplates containing 200 μl of half-strength Murashige and Skoog (MS) medium (2.2 g l$^{-1}$ MS salts, 5 g l$^{-1}$ sucrose, pH 5.7 to 5.8 and 1 g l$^{-1}$ agar) supplemented with one of the selected 2,000 diverse chemicals at a concentration of 50 to 100 μM (usually dissolved in DMSO). For phenotype analysis or ethylene biosynthesis quantification, surface-sterilized seeds were sown on MS medium (4.4 g l$^{-1}$ MS salts, 10 g l$^{-1}$ sucrose, pH 5.7 to 5.8 and 8 g l$^{-1}$ agar) supplemented with the indicated concentrations of ACC, and/or PZA/POA, and imbibed at 4 °C for 3 days. For etiolated seedlings, plates were kept under light for 3 to 4 h after imbibition, and then incubated in the dark at 22 °C for 3 days. For green seedlings, the plates were placed at 22 °C with a 16L/8D illumination cycle after imbibition. For *Arabidopsis* suspension cultured cell PSB-D (ecotype Landsberg erecta)[63], cells are grown in 250 ml flasks at 27 °C in light-protected shakers at 130 r.p.m., and subcultured in fresh medium (4.3 g l$^{-1}$ MS Salt, 100 mg l$^{-1}$ myo-inositol, 0.4 mg l$^{-1}$ thiamine hydrochloride (Vitamin B1), 50 mg l$^{-1}$ kinetin, 800 mg l$^{-1}$ 1-naphthaleneacetic acid and 30 g l$^{-1}$ sucrose, pH 5.7) once per week by transferring 5 ml of old cells into 45 ml of fresh medium.

**Chemical solutions.** The small molecule library (SP 2,000, http://www.msdiscovery.com) screen was performed at the University of California at Riverside with a stock concentration of 5 μg ml$^{-1}$ in DMSO. Other chemicals used in this work were purchased from Sigma-Aldrich. The stock solutions were prepared at the concentration indicated: PZA (100 mM), POA (100 mM), 2-PA (100 mM) and ACC (10 mM). ACC was dissolved in water and filtered with 0.22 μm sterilized filters, whereas other chemicals were dissolved in DMSO.

**Measurement of ethylene production.** Ethylene production was determined as follows: 30 3-day-old etiolated or 5-day-old green seedlings of Col-0, *eto1-2* or other mutants were selected randomly and sealed in a 30 ml vial containing 3 ml liquid MS medium supplemented with indicated treatments. The vial was sealed with a rubber stopper for 24 h at 22 °C prior to ethylene sampling. A 250 μl sample of the headspace gas was withdrawn using a gas-tight syringe from each chamber through the septum stopper, and injected into a gas chromatograph (Agilent 6890NGC, Agilent Technologies, USA) that was equipped with a HP-PLOTQ column (40 m × 530 μm × 40 μm) and a flame ionization detector (FID)[44]. Separations were carried out at 60 °C using N$_2$ as the carrier gas. The area of the ethylene peak was integrated with Agilent Chemstation and results were expressed as the average relative ethylene production (%) of each treatment.

**Quantification of ACC and M-ACC.** 10-day-old green seedlings of *eto1-2* under different treatments were ground in liquid nitrogen, and extracted with 5% SSA (sulfosalicylic acid). For ACC measurement, the extract was oxidized with 2 mM HgCl$_2$, 0.33% NaOCl and 0.2 M NaOH, reacted on ice for 4 min, and then the ethylene production was determined with GC-FID. For M-ACC measurement, the extract was hydrolyzed using 2 M HCl followed by oxidization and ethylene production determination, and the content of ACC and M-ACC were calculated based on the ethylene production and efficiency of oxidization.

**EIN3 protein level quantification.** Total plant proteins were extracted using extraction buffer (50 mM Tris-Cl, pH 7.5, 1 M NaCl, 10% glycerol (v/v), 0.1% Tween 20 (v/v), 1 mM DTT and 1 × protease inhibitor cocktail (Roche)) after grinding in liquid nitrogen, and boiled with 2× SDS sample buffer (0.1 M Tris-HCl, pH 6.8, 4% SDS (w/v), 20% glycerol (v/v), 2% β-mercaptoethanol (v/v), 1 mM DTT and 0.02% bromophenol blue (w/v)) before loading on a 10% SDS-PAGE gel for separation. Western blotting was performed following standard procedures using an anti-GFP antibody (Abcam, ab13970) with a 5,000 times dilution.

**Histochemical β-glucuronidase staining.** Seedlings were grown for three days on the indicated medium in the dark, then collected and washed with phosphate buffer saline and stained with β-Glucuronidase staining buffer (50 mM sodium phosphate buffer, pH 7.0, 10 mM Na$_2$EDTA, 0.5 mM K$_4$[Fe(CN)$_6$] · 3H$_2$O, 0.5 mM K$_3$[Fe(CN)$_6$], 0.1% Triton X-100 (v/v) and 1 mg ml$^{-1}$ X-Gluc). The staining reaction was terminated with 70% ethanol, and the seedlings were mounted on slides in Hoyer's solution (chloral hydrate:water:glycerol; 8:3:1; w/v/v) and examined by microscopy. Each treatment was carried out with ten seedlings and one representative picture is shown.

**Protein expression and purification.** AtACO2$^{1-303}$ was expressed in *E. coli* BL21(DE3) using a modified pQlink vector with an N-terminal GST-tag followed by a tobacco etch virus protease cleavage site[64]. For protein expression, cultures were grown at 37 °C in LB medium to an OD$_{600}$ of 0.8 before induced with 0.5 mM IPTG overnight at 18 °C. Cells were collected by centrifugation and frozen at −80 °C.

GST-AtACO2$^{1-303}$ was first purified from cell lysates using glutathione sepharose 4B (GE Healthcare) affinity chromatography. 25 mM EDTA was added to the lysis buffer (25 mM Tric-HCl, pH 8.0, 300 mM NaCl, 1 mM DTT, 1 mM PMSF) and GST elute buffer (50 mM Tric-HCl, pH 8.0, 300 mM NaCl, 1 mM DTT, 15 mM glutathione) to chelate metal ions that may bind to AtACO2 from *E. coli*. After digestion by tobacco etch virus protease, the GST tag and uncleaved proteins were removed by a second GST affinity chromatography step. Untagged AtACO2$^{1-303}$ was further purified using size-exclusion chromatography (Superdex 200 column; GE Healthcare) with gel filtration buffer (20 mM Bis–Tris, pH 6.0, 50 mM NaCl, 1 mM DTT). Purified AtACO2$^{1-303}$ was concentrated and flash-frozen with liquid nitrogen.

AtACO2 mutants were purified as described above. GA2OX2, ANS and DAO were purified similar to AtACO2 without the size-exclusion chromatography step.

**ACO enzymatic activity assay and kinetic analysis.** The activity was performed following previously published protocol[45]. The reaction mixture contains 50 mM MOPS, pH 7.4, 10% glycerol (v/v), 5 mM ascorbic acid, 20 mM NaHCO$_3$, 0.02 mM FeSO$_4$, 1 mM DTT, and different amounts of ACC and PZA/POA as indicated. The reaction was initiated by the addition of 5 μg ACO protein and immediately sealed with a rubber stopper. After incubation at 30 °C for 30 min with shaking, 100 μl of gas product was withdrawn by syringe from the headspace of the vial and injected

into the GC-FID. The area of the ethylene peak was integrated with Agilent Chemstation and results were analysed using the program GraphPad Prism (version 5.01; GraphPad Software Inc., San Diego, CA, USA). The initial velocities were plotted against starting ACC concentrations in a double-reciprocal manner and the kinetic parameters were deduced.

**The Wayne test.** 500 μM PZA were added to 50 ml *Arabidopsis* suspension cell log phase cultures or boiled cultures and incubated at 27 °C for 24 h. 1 ml cell culture was collected after centrifugation, and 10 μl 2% ferrous sulfate was added to the supernatant for colour development. The absorbance at 450 nm of each samples was measured by spectrophotometer and the POA concentration was calculated according to the POA-Abs$_{450}$ standard curve drawn using the reference solutions of POA.

**Mass spectrometry.** *Arabidopsis* suspension cells were cultured in liquid MS medium in the presence or absence of 500 μM PZA for 24 h. The culture supernatant was collected by centrifugation and alkalized (pH > 11) using sodium hydroxide (NaOH), followed by extracting with ethyl acetate twice to eliminate lipid-soluble molecules. The aqueous phase was collected and acidified (pH < 2) using hydrochloric acid (HCl). The protonated cultures were extracted with ethyl acetate five times to enrich POA. The resulting samples were concentrated by reduced pressure distillation and analysed using a Fourier Transform Ion Cyclotron Resonance Mass Spectrometer (Solarix XR, Bruker).

**PZase assay and nuclear magnetic resonance analysis.** GST-tagged AtNIC2 and AtNIC3 were purified from *E. coli* using the glutathione sepharose 4B (GE Healthcare) affinity method similarly as described above for GST-AtACO2$^{1-303}$, and the PZase assay was performed in the reaction buffer (25 mM Tris-HCl, pH 7.5, 150 mM NaCl, 1 mM MgCl$_2$ and 5 mM PZA) for 12 h at 30 °C (ref. 65), followed by TLC separation. Silica TLC plates were developed in dichloromethane/methanol (3:1), and analysed with ultraviolet.

For NMR analysis, the AtNIC2-catalysed PZase reaction was acidified using HCl and extracted using ethyl acetate. The solvent was removed by reduced pressure distillation. The sample was then re-dissolved in dimethyl sulfoxide-d6 and analysed using Bruker-400 MHz NMR (ARX400, Bruker).

**Fragmentation of AtACO2.** Purified AtACO2 was incubated with or without 200 μM POA at 30 °C in a buffer containing 50 mM MOPS, pH 7.2, 10% glycerol (v/v), 5 mM ascorbic acid, 20 mM NaHCO$_3$, 0.02 mM FeSO$_4$, 1 mM ACC and 1 mM DTT. At the indicated time point, the reaction was terminated by the addition of SDS-PAGE loading buffer followed by boiling. Fragmentation of AtACO2 was visualized on a SDS-PAGE gel.

**Isothermal titration calorimetry.** ITC experiments were performed using a MicroCal iTC200 (GE Healthcare) at 25 °C. Small molecules and proteins were in the same buffer containing 25 mM Bis–Tris (pH 6.0), 150 mM NaCl, 0.1 mM ZnCl$_2$ (or 0.2 mM FeSO$_4$ for POA and AtACO2). The concentration of small molecules in the syringe was 1 mM, and the concentration of proteins in the cell was 90 μM. Data were analysed with a one-site model using Origin 7.0 (OriginLab).

**Confocal laser microscopy.** GFP fluorescence and propidium iodide staining signals in roots were detected using Zeiss LSM-710 microscope. All seedlings were grown in the dark for 3 days on MS medium supplemented with the indicated treatments, and then mounted on glass slides and observed in water. For propidium iodide staining, roots were visualized in the presence of propidium iodide without fixation. ImageJ was used to process images and to quantify the fluorescent intensity of GFP.

**Anthocyanin measurement.** 6-day-old green seedlings grown on MS medium supplemented with different concentration of PZA were sampled for anthocyanin measurement by spectrophotometry[66]. The content of anthocyanin is presented as $(A_{535} - A_{650})/g$.

**Crystallization and data collection.** For the crystallization of AtACO2$^{1-303}$-POA, purified AtACO2$^{1-303}$ was concentrated to 9 mg ml$^{-1}$ in a buffer containing 20 mM Bis–Tris (pH 6.0), 50 mM NaCl, 1 mM DTT, 2 mM ZnCl$_2$ and 20 mM POA. The AtACO2$^{1-303}$-POA crystals were grown at 18 °C by the sitting-drop vapour diffusion method, using a 1:1 ratio of protein:reservoir solution containing 21.5–24% PEG 3350 and 100 mM citric acid (pH 5.0). Crystals grew to full size in several days and were transferred to 24% PEG 3350, 100 mM citric acid (pH 5.0), 20 mM POA, 2 mM ZnCl$_2$ and 14% ethylene glycol before being flash-frozen under liquid nitrogen. For AtACO2$^{1-303}$-2-PA crystallization, purified ACO2$^{1-303}$ was concentrated to 9 mg ml$^{-1}$ in a buffer of 25 mM Bis–Tris (pH 6.0), 50 mM NaCl, 1.5 mM ZnCl$_2$ and 5 mM 2-PA. The ACO2$^{1-303}$-2-PA crystals were grown at 18 °C by the sitting-drop vapour diffusion method, using a 1:1 ratio of protein:reservoir solution containing 30–32% PEG 3350 and 100 mM citric acid

(pH 5.3). Crystals grew to full size in several days and were transferred to 32% PEG 3350, 100 mM citric acid (pH 5.3), 5 mM 2-PA, 1.5 mM ZnCl$_2$ and 4% ethylene glycol before being flash-frozen under liquid nitrogen.

AtACO2$^{1-303}$-POA and AtACO2$^{1-303}$-2-PA diffraction data sets were collected at Shanghai Synchrotron Radiation Facility (SSRF) beamline BL17U and BL18U, respectively. The diffraction data were indexed, integrated and scaled using the program HKL2000 (HKL Research).

**Structure determination and refinement.** The AtACO2–POA structure was determined by molecular replacement using the program Phaser[67]. The published PhACO structure (PDB ID: 1WA6) was used as the search model. Initial model building was carried out with the Autobuild program in Phenix[68]. The structural model was then adjusted using Coot[69] and refined using Phenix.refine[68]. The AtACO2-2-PA structure was determined by molecular replacement using the AtACO2 structure obtained above as the search model, and refined using Phenix.refine. The refinement quality of two structures was checked using MolProbity[70]. All residues are located in the favoured or allowed regions of the Ramachandran plot.

**Statistical analyses.** The quantitative values obtained in the figures were analysed in Excel spreadsheets with the embedded basic statistical functions (mean, s.d., Student's *t*-test, linear regression).

**Data availability.** Coordinates and structure factors have been deposited in the Protein Data Bank under accession codes of 5GJ9 and 5GJA for the structures of AtACO2–POA and AtACO2–2-PA, respectively. All other data supporting the findings of this work are available from the corresponding authors upon reasonable request.

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

## Acknowledgements

We thank staff members of beamline BL17U and BL18U at SSRF (Shanghai Synchrotron Radiation Facility, China) for assistance in data collection, the Core Facilities at School of Life Sciences, Peking University for assistance with the ITC instrument, and Professors Xuefeng Fu, Zikuan Wang, Yuntao Zhu and Zhen Sun at College of Chemistry and Molecular Engineering, Peking University for discussions. This work is supported by National Natural Science Foundation of China (91217305 and 91017010) and start-up funding from Southern University of Science and Technology (Y01226124) to H.G., National Natural Science Foundation of China (31570735) and National Key Research & Development Plan (2016YFC0906000) to J.X., and the Peking-Tsinghua Center for Life Sciences to J.X. and H.G.

## Author contributions

X.S., Y.L., J.X. and H.G. designed the experiments. X.S., W.H., P.X. and Y.W. performed the biological experiments, biochemical assays and the characterization of small molecules. Y.L., C.J. and S.D. performed the crystal structure study. H.L. and N.R. conducted the chemical library screens. X.S., Y.L., J.X. and H.G. wrote the manuscript.

## Additional information

**Competing interests:** The authors declare no competing financial interests.

**Publisher's note**: 

