## [Peer Review File · Nature Communications]

This manuscript has been previously reviewed at another journal that is not operating a transparent peer review scheme. This document only contains reviewer comments and rebuttal letters for versions considered at Nature Communications. Mentions of prior referee reports have been redacted.

Reviewers' Comments:

Reviewer #1 (Remarks to the Author)

This is a very interesting manuscript where the authors have shown that Pyrazinamide (PZA), an antimicrobial agent which plays an important role in shortening tuberculosis therapy, acts as an inhibitor of ethylene biosynthesis by decreasing ACO activity in Arabidopsis. The authors suggest that the conversion of PZA to pyrazinoic acid (POA), the active derivative of PZA, can be mediated by the enzymatic activity of Arabidopsis PZase/nicotinamidase AtNIC1, AtNIC2, and AtNIC3. Furthermore, they propose the inhibitory mechanism of POA on ACO activity by structural analyses.

The data in this manuscript are presented in a mostly well-structured manner and have provided enough evidence to show specificity of the inhibitor in plants. There are some minor points that the authors need to address.

1) There is a discrepancy between Fig 1C and Fig 2A regarding the root length of Col-0 upon treatment of PZA. As PZA decreases ethylene production by inhibiting ACO activity, the root length of etiolated seedlings is expected to be reduced. This was presented in Fig 1C, but not in Fig A. Did the Authors use the lower concentration of PZA in the Fig. 2A than the experiment in the Fig. 1C?

2) Several studies including recent paper published in Plant Physiology (Street IH et al., 2016) showed that ethylene has a role in cell division in the root meristem. However, the authors claimed that PZA does not affect root meristem size of eto1 mutant. The authors need to discuss this difference in relation to the role of PZA.

3) would like to see the sequence similarity and domain conservation of PZase/nicotinamidase between plant and animal system.

Reviewer #2 (Remarks to the Author)

The manuscript describes identification of PZA derivatives as inhibitors of ethylene biosynthesis in Arabidopsis. Insights from a phenotype-based chemical biology approach are extended by auspicious examination of binding/ activity assays using a reasonably well-characterised panel of isolated plant enzymes and in cultured cells. One key finding is the identification of PZA as pro-drug and its metabolite in Arabidopsis, POA as the active inhibitor of ACC oxidases. In addition, the authors determined crystal structures of their target enzyme, ACO2, which provided insights into mode of inhibition by these compounds.

One criticism is the quality of the MS data the authors provided for characterisation of POA in the cultured cells (Supplementary Figure 6). The intensity of the peak for POA (at m/z 123.02) is almost at the background in the top panel with poor signal-to-noise ratio. Would it be possible to repeat the experiment using LC-MS? A separation step prior to MS will perhaps enrich the signal. The figure needs more discussion/ clarification in the legend.

Please provide the buffer conditions used for the purification of enzymes (at least for the final-step purification).

Reviewer #3 (Remarks to the Author)

The manuscript by Sun et al describes the identification of an inhibitor of ACC Oxidase enzyme in Arabidopsis thaliana, by way of a chemical genetics screen. The authors properly demonstrate through a series of detailed and well controlled experiments that this new compound, PZA, and its

derivative, POA, are potent inhibitors of ACO activity, having an effect on all ACO enzymes in Arabidopsis, and thus may be used as a broad ethylene inhibitor, with potential applications in plant biology research and agriculture.

The manuscript is well written and the experiments are well performed and have the appropriate controls. The scientific contribution is between moderate and high. The authors have identified a new inhibitor, but its identification and mode of action do not reveal any new insights about the regulation of ethylene biosynthesis in plants, or about the structures/activity of the ACO enzymes. The major contribution is the potential use of this compound in agriculture and plant biology research. In this regard, the well determined mode of action of PZA could aid in the identification/engineering of even more potent ACO inhibitors.

Minor comments are as follows:

- The ethylene pathway could be better explained (in more detail) in the Introduction
- Line 122: when the authors write "promoted cell length", do you mean promote cell expansion?
- Line 183/284: "by using", change to either by or using.
- The experiment shown in figure 1 is poorly explained in the text.
 - Fig 2A: PZA seems to make hypocotyls longer in Col, even without ethylene, indicating an inhibition of ACO and decrease in basal ethylene levels. This should be noted.
 - Fig 4B: To make the figure easier to read, it would be better to add "POA" on the graph legend next to the concentrations used
 - Fig 4c: To make the figure easier to read, it would be better to add the concentration of POA used in the graph
 - Fig 4D: These graphs are very small, they will be harder to read when in print, consider making them bigger
 - Fig 5B: the compounds used should be named in the text and in the legend
- Methods: Mention which mutant alleles used, and cite their references

Reviewer #1:

This is a very interesting manuscript where the authors have shown that Pyrazinamide (PZA), an antimicrobial agent which plays an important role in shortening tuberculosis therapy, acts as an inhibitor of ethylene biosynthesis by decreasing ACO activity in Arabidopsis. The authors suggest that the conversion of PZA to pyrazinoic acid (POA), the active derivative of PZA, can be mediated by the enzymatic activity of Arabidopsis PZase/nicotinamidase AtNIC1, AtNIC2, and AtNIC3. Furthermore, they propose the inhibitory mechanism of POA on ACO activity by structural analyses.

The data in this manuscript are presented in a mostly well-structured manner and have provided enough evidence to show specificity of the inhibitor in plants. There are some minor points that the authors need to address.

1) There is a discrepancy between Fig 1C and Fig 2A regarding the root length of Col-0 upon treatment of PZA. As PZA decreases ethylene production by inhibiting ACO activity, the root length of etiolated seedlings is expected to be reduced. This was presented in Fig 1C, but not in Fig A. Did the Authors use the lower concentration of PZA in the Fig. 2A than the experiment in the Fig. 1C?

Author's response: We thank the reviewer for this critical comment. The slight discrepancy between Fig. 1C and Fig. 2A does exist and the reason for this is probably due to the different growth condition in this two treatments. In Fig. 2A, the medium plates were placed vertically and the root grew on the medium surface, however, in Fig. 1C, the medium plates were placed horizontally and the root grew into the medium, where roots face more mechanical resistance, and likely produce more ethylene (Biddington, Norman L. "The effects of mechanically-induced stress in plants—a review." *Plant Growth Regulation* 4 (1986): 103-123). Another possible reason is that the two different growth conditions led to differential PZA absorption by roots, wherein roots on vertically plated medium had smaller contact surface area with MS medium and likely had less absorption. Together, in Fig. 2A, roots might produce little ethylene while absorb less PZA, so the effect of PZA's inhibition on ethylene production is slighter. However, the effect of PZA on root growth and ethylene production of *eto2* or ACC-treated Col is quite pronounced regardless of growth conditions. So we used *eto2* or Col treated with ACC to demonstrate the effect of PZA, POA or their derivatives during the rest of study.

*2) Several studies including recent paper published in Plant Physiology (Street IH et al., 2016) showed that ethylene has a role in cell division in the root meristem. However, the authors claimed that PZA does not affect root meristem size of *eto1* mutant. The authors need to discuss this difference in relation to the role of PZA.*

Response: We appreciate the reviewer for this valuable suggestion. Ethylene inhibits

root growth of *Arabidopsis* mainly by regulating root cell elongation, but it is still controversial whether ethylene impacts meristem activity. For example, the above-mentioned *Plant Physiology* paper showed that ethylene has a role in cell division in the root meristem, in which high dose of ACC/ethylene treatment on Col green seedlings for 7 days led to reduced SAM size. Interestingly, they also showed that the SAM sizes of Col, *ein2* and *ein3eil1* mutants are comparable. By contrast, other papers suggest that ethylene inhibits root growth by regulating root cell elongation without impacting meristem activity (Růžička, Kamil, et al. "Ethylene regulates root growth through effects on auxin biosynthesis and transport-dependent auxin distribution." *The Plant Cell* 19 (2007): 2197-2212; Mao, Jie-Li, et al. "Arabidopsis ERF1 mediates cross-talk between ethylene and auxin biosynthesis during primary root elongation by regulating ASA1 expression." *PLoS Genet* 12 (2016): e1005760). A possible explanation for such inconsistency is that growth conditions and measurement method used in those papers are different. In the *Plant Physiology* paper, the SAM size was determined based on the exact cell number of cortex cells in a file extending from the QC to the first elongated cell, while in other papers and our data, the length of the SAM is measured directly. To avoid this controversy, we have rephrased our results and conclusion in our revised manuscript.

3) would like to see the sequence similarity and domain conservation of PZase/nicotinamidase between plant and animal system.

Response: *Mycobacterium tuberculosis* nicotinamidase/pyrazinamidase (MtPZase) converts the anti-tuberculosis drug PZA to the active form POA. The three AtNICs (AtNIC1, AtNIC2, and AtNIC3) show limited sequence similarity with the MtPZase (27%, 34%, 25% identity, respectively). It appears that the PZases display great sequence diversity while maintaining the same general fold and function. In fact, the three AtNICs are only 15-16% identical to the yeast nicotinamidase PNC1, and AtNIC1 is only 27% and 13% identical to AtNIC2 and AtNIC3, respectively (Wang et al., "Nicotinamidase participates in the salvage pathway of NAD biosynthesis in Arabidopsis." *Plant J.* 49(6):1020-9.). Nevertheless, AtNIC1 clearly displayed nicotinamidase activity (Wang et al.), and we showed AtNIC2 and AtNIC3 can convert PZA to POA (**Fig. 3C**). The crystal structure of MtPZase in complex with an Fe²⁺ ion has been determined (Petrella, S, et al. "Crystal Structure of the Pyrazinamidase of Mycobacterium tuberculosis: Insights into Natural and Acquired Resistance to Pyrazinamide." *Plos One* 6 (2011):e15785.). The Fe²⁺ ion is important for the enzyme to bind to the substrate nicotinamide or PZA molecule, as suggested by a previous study on a homolog protein from *Acinetobacter baumannii* (PK Fyfe et al., "Specificity and Mechanism of *Acinetobacter baumannii* Nicotinamidase: Implications for Activation of the Front-Line Tuberculosis Drug Pyrazinamide", *Angew. Chem. Int. Ed.* 48, 9176-9179). In the MtPZase structure, the Fe²⁺ ion is coordinated by Asp49 and three histidines (His51, His57, His71). Furthermore, the MtPZase was proposed to use a Cys138-Asp8-Lys96 motif evocating a cysteine-based catalysis. No crystal structure information is available for the AtNICs yet.

Nevertheless, when the structural model of AtNIC1 (SWISS-MODEL database ID: Q8S8F9) is compared to the structure of MtPZase, residues critical for substrate-binding and catalysis appear to be mostly present (see the figure below, **Panel a**). In AtNIC1, Cys167-Asp37-Lys127 likely functions similarly as the Cys138-Asp8-Lys96 catalytic motif in MtPzase. Asp85, His87, and His99 in AtNIC1 may be responsible for binding to the divalent metal ion, similarly as Asp49, His51, and His71 in the MtPZase. Though a residue corresponding to MtPZase His57 is absent in AtNIC1, an aspartate (Asp128) is located in the vicinity and may have an analogous function. These critical residues are also present in AtNIC2 and AtNIC3, when their structural models (SWISS-MODEL database ID: Q9FMX7 and Q9FMX8) were examined (**Panel b**). These observations further support previous and our biochemical results, suggesting that the AtNICs can function as *bona fide* nicotinamidases/pyrazinamidases.

Alignments between AtNICs and MtPZase

(a) Superimposition of the AtNIC1 structural model (limegreen) and the crystal structure of MtPZase (pink).

(b) Sequence alignment of the three AtNICs. Residues that may be responsible for metal ion binding are highlighted with yellow boxes. Residues corresponding to the MtPZase catalytic motif are highlighted with blue boxes.

Reviewer #2:

The manuscript describes identification of PZA derivatives as inhibitors of ethylene biosynthesis in Arabidopsis. Insights from a phenotype-based chemical biology approach are extended by auspicious examination of binding/ activity assays using a reasonably well-characterised panel of isolated plant enzymes and in cultured cells. One key finding is the identification of PZA as pro-drug and its metabolite in Arabidopsis, POA as the active inhibitor of ACC oxidases. In addition, the authors determined crystal structures of their target enzyme, ACO2, which provided insights into mode of inhibition by these compounds.

One criticism is the quality of the MS data the authors provided for characterisation of POA in the cultured cells (Supplementary Figure 6). The intensity of the peak for

POA (at m/z 123.02) is almost at the background in the top panel with poor signal-to-noise ratio. Would it be possible to repeat the experiment using LC-MS? A separation step prior to MS will perhaps enrich the signal. The figure needs more discussion/ clarification in the legend.

Response: We thank the reviewer for this excellent suggestion. In the repeated experiment, we greatly scaled up the starting materials, treated the culture supernatant with NaOH (pH > 11) followed by extracting with ethyl acetate twice to eliminate lipid-soluble molecules. The aqueous phase was collected and treated with HCl (pH < 2), and protonated cultures were extracted with ethyl acetate five times to enrich POA. The resulting samples were concentrated by reduced pressure distillation, and subjected to MS analysis. In the new data, the POA peak is much more pronounced compared with the background peaks shown in the revised manuscript (Supplementary Fig. 6). We also modified the legend of Supplementary Fig. 6 and described the new methods to extract POA in the Methods.

Please provide the buffer conditions used for the purification of enzymes (at least for the final-step purification).

Response: We have included details of buffers used for the purification of enzymes in the **Methods** of the revised manuscript (Page 11, Lines 426, 427, 432).

Reviewer #3:

The manuscript by Sun et al describes the identification of an inhibitor of ACC Oxidase enzyme in Arabidopsis thaliana, by way of a chemical genetics screen. The authors properly demonstrate through a series of detailed and well controlled experiments that this new compound, PZA, and its derivative, POA, are potent inhibitors of ACO activity, having an effect on all ACO enzymes in Arabidopsis, and thus may be used as a broad ethylene inhibitor, with potential applications in plant biology research and agriculture.

The manuscript is well written and the experiments are well performed and have the appropriate controls. The scientific contribution is between moderate and high. The authors have identified a new inhibitor, but its identification and mode of action do not reveal any new insights about the regulation of ethylene biosynthesis in plants, or about the structures/activity of the ACO enzymes. The major contribution is the potential use of this compound in agriculture and plant biology research. In this regard, the well determined mode of action of PZA could aid in the identification/engineering of even more potent ACO inhibitors.

Minor comments are as follows:

-The ethylene pathway could be better explained (in more detail) in the Introduction

Response: As suggested, we have expanded the description on the ethylene signaling pathway in the revised manuscript (Page 1 & 2, Lines 36-44).

-Line 122: when the authors write "promoted cell length", do you mean promote cell expansion?

Response: Yes. Our data showed that PZA treatment made root longer by mainly promoting cell elongation/expansion in the root maturation zone.

-Line 183/284: "by using", change to either by or using.

Response: it has been revised.

-The experiment shown in figure 1 is poorly explained in the text.

Response: Accordingly, we add more descriptions for the contents of Fig. 1 in the revised manuscript (Page 4, Lines 123-128).

-Fig 2A: PZA seems to make hypocotyls longer in Col, even without ethylene, indicating an inhibition of ACO and decrease in basal ethylene levels. This should be noted.

Response: We really appreciate this reminder and add the description for this observation accordingly.

-Fig 4B: To make the figure easier to read, it would be better to add "POA" on the graph legend next to the concentrations used

-Fig 4c: To make the figure easier to read, it would be better to add the concentration of POA used in the graph

-Fig 4D: These graphs are very small, they will be harder to read when in print, consider making them bigger

Response: We appreciate these suggestions and have made changes accordingly.

-Fig 5B: the compounds used should be named in the text and in the legend

Response: We have named the compounds in the legend of Fig. 5 and Supplementary Fig. 10 in the revised manuscript.

-Methods: Mention which mutant alleles used, and cite their references

Response: We have listed all the mutant alleles used in the paper, and cited proper references in the revised manuscript.

Reviewers' Comments:

Reviewer #1:

Remarks to the Author:

The authors adequately addressed all the questions in the review comments either by presenting new data or by adding more discussion.

Reviewer #2:

Remarks to the Author:

The authors have sufficiently addressed to my previous comments.

Can you please include an MS spectrum for PZA reference in the Supplementary Figure 6 and the structures of both POA and PZA on their respective spectra (similar to Figure 3D) along with their calculated high-res masses underneath? I think this will be useful to the readers.

Reviewer #1:

The authors adequately addressed all the questions in the review comments either by presenting new data or by adding more discussion.

Author's response: We thank the reviewer for the positive evaluation and kind help in improving our manuscript.

Reviewer #2:

The authors have sufficiently addressed to my previous comments.

Can you please include an MS spectrum for PZA reference in the Supplementary Figure 6 and the structures of both POA and PZA on their respective spectra (similar to Figure 3D) along with their calculated high-res masses underneath? I think this will be useful to the readers.

Author's response: Thanks again for all the valuable comments. As suggested, we add the MS spectra of PZA and POA as well as their structures and high-resolution masses in the supplementary Figure 6 of the revised manuscript.